# Survival of musical activities. When do young people stop making music?

**Nicolas Ruth** *, **Daniel Müllensiefen**

Department of Psychology, Goldsmiths, University of London, New Cross, London, United Kingdom

* n.ruth@gold.ac.uk

## Abstract

Although making music is a popular leisure activity for children and adolescents, few stay musically engaged. Previous research has focused on finding reasons for quitting musical activities, pedagogical strategies to keep students engaged with music, and motivational factors of musical training. Nonetheless, we know very little about how the proportion of musically active children changes with age and what traits influence the survival of musical engagement. This study used longitudinal data from secondary school students in the UK and Germany aged between 10 and 17 years. A survival analysis was applied to investigate the trajectories of musical activities across this age span. Other factors like type of learned instrument, gender, personality and intelligence were taken into account for further analyses using generalized linear models. Results indicate that about 50% of all students drop out of music lessons and other musical activities by the time they turn 17 years old, with most students quitting between the ages of 15 and 17. Musical home environment is an important factor that is associated with lower drop out rates while conscientiousness and theory of musicality showed smaller significant associations.

## 1. Introduction

Why did I ever stop making music, is a question that probably many people asked themselves at some point of time (if they ever engaged in any musical activity). Typically, people with a middle class background in western societies start learning an instrument and join an orchestra or band at 8–12 years old [1], but many children are already enthusiastic about learning an instrument at the age of 6 or 7 [2]. While there are some people who keep engaging with music for many years, many more stop practising music at some point during childhood or adolescence [3]. We know that music is a popular leisure activity for children and adolescents [4], but not all of them actually engage in musical lessons or activities and even fewer 'survive' musically–which in this case means staying engaged in their musical activities while growing up.

Many researchers have investigated children dropping out of musical activities [5] to predict the age they will stop taking part and work out the reasons behind this. This line of research goes back to the 1960s [6] and various reasons for quitting musical activities have been identified [3] but apart from very few exceptions [7], there is a lack of longitudinal data

files are available from the Goldsmiths' Department of Psychology, psychology@gold.ac.uk.

**Funding:** This research has been supported by the Humboldt's foundation's Feodor-Lynen post-doctoral fellowship for Nicolas Ruth and the Anneliese-Maier research prize awarded to Daniel Müllensiefen by the Humboldt foundation.

**Competing interests:** The authors have declared that no competing interests exist.

musical engagement and dropouts. Therefore, this paper aims to apply a rigorous empirical model to longitudinal data in order to answer the questions of who will stay engaged in musical activities and what factors can predict who is more likely to stop, as well as determine the typical length of the period in which children continue to make music.

## 1.1 Musical engagement

Engaging in musical activities of any kind can have positive consequences in many areas (for an overview see [8]). For instance, musical engagement and shaping musical abilities are important for the development of various traits and skills in children and adolescents [9–12]. Making music with others helps young people in terms of identity development and also helps them cope with issues that occur while growing up [13,14].

These positive effects are a major motivation for parents, music instructors and school-teachers to encourage continuous engagement of their children and students. The role of parents and their support is crucial for children's motivation. The association between parental support, and the length of time spent playing music has been investigated by surveying a group of piano students and their parents [15] and groups of violin students [16]. Although parents seem to be the most important factor for children's musical engagement, the study by Creech and Hallam [17] showed that the relationship with their music teachers also strongly contributes to continued music making.

In addition, several individual factors have been identified that drive musical development. For example, physical predispositions, personality, emotions towards music, and gender [18] influence how individuals engage with music. In addition, acquired self-regulatory strategies can partly explain why some people engage with music and some do not [7]. When asked in a representative randomized survey (with about 11,000 participants of all ages from Germany) about the reasons to become musically active, the most common answers were "fun, being creative, and doing something meaningful for leisure" [19].

The previous findings support the results of one of the few longitudinal studies on development of musical engagement by McPherson and colleagues [5], who conclude that the four most important factors for children and adolescents to learn an instrument are: The expectation of being capable to learn an instrument; the enjoyment of playing an instrument; acquiring self-regulatory strategies for learning the instrument in question; and the support of family members, music instructors and friends.

But sometimes children are not supported or do not feel motivated for various reasons [20]. Thus, a lot of research is dedicated to investigating why people stop making music.

## 1.2 Reasons for dropping musical activities

For children and adolescents, engaging with a range of different leisure activities is fairly common [21]. Some of them are kept while others are dropped. Often children and adolescents take up sport, gaming, or music activities when they start going to school, changing between them and sometimes quitting very soon after when they enter a new life phase [5]. Taking a closer look at musical trajectories, we often find that musicians quit making music in the transition from adolescence to adulthood, usually when they start working or go to a university [3,22,23].

In a longitudinal study with 157 music school students from Australia, all between 7 and 22 years old, McPherson and colleagues [5] found that children also drop out during the transition from primary to secondary school. In secondary school musical engagement drops very quickly from first years to seniors. The stages of musical learning are often discussed and are usually defined by the proportion of students quitting music [18]. A proposed cluster of age

groups is 9–12, 13–17, and 18–24, which can be ranked from very musically active (first group) to not very musically active (last group [23]).

The factors influencing premature dropouts have been reviewed to offer suggestions that may help teachers prevent losing students [24]. The most common reasons are socio-economic factors, loss of motivation, loss of support and logistical issues for example with transport to or from the music school [25]. Another reason for dropping out that was identified from data of a survey of 120 music teachers from Queensland, Australia was the perceived quality of the music programme [26]. In addition, an interview study with five members of an orchestra and a big band found that the social aspects of playing music and the interaction with other music students were major reasons for continuing with or dropping out of musical activities [27]. A retrospective survey of over 3,000 adults from the Swedish Twin Registry identified several factors that were important predictors for a continued playing of musical instruments or singing, like starting age of studying music, the instrument played, type of teaching, content, number, and characteristcs of lessons, the musical environment during childhood and the preferred music genre [28].

Several studies indicated that psychological factors such as personality traits and cognitive or affective abilities are involved in the decision to stop playing music. In a meta review Hallam [18] states that a person's personality was considered to contribute to an individual's motivation which is closely tied to the probability of staying musically active. In addition, students with a higher IQ were found to be more likely to engage with music lessons in a positive way (correlations between amount of music lessons and two different intelligence measures were between $r$ = .35-.36 [29]). Also, demographics like age and gender were good predictors for musical dropouts as indicated by the analysis of Switlick and Bullerjahn [3] that showed in a descriptive way that children between 11 and 15 years tend to drop out of music lessons more frequently compared to other age groups. The authors also report that girls and boys differ in their reasons for quitting music lessons–girls more often state that they lose motivation while boys claim more frequently to have reached their learning goals. The same study [3] also showed that reasons for drop out differed between players of different instruments, with guitar players stating that they reached their learning goals more often than pupils learning other instruments. Nevertheless, a positive result from this research strand on musical drop out is that people who quit instrumental lessons usually value the time they spent making music and consider it enriching despite having stopped making music [30].

## 1.3 Survival analysis

To investigate the trajectories of musical activity, longitudinal data and suitable statistical data analysis methods are needed. One promising technique is survival analyses which is often used to investigate the survival of people with diseases [31]. There are several studies which have used survival analysis in a musical context. For example, Byrgen and colleagues [32] showed how various cultural and music related factors help people aged between 16 and 74 years to stay alive. In this study, 6301 men and 6374 women from Sweden were followed and of them 533 men and 314 died during the time of study from 1982 to 1983. Their results show that the investigated control variables, namely attendance at cultural events, reading, listening to music and making music, were associated significantly with well-being.

Other survival analyses with a connection to music come from economics. For example, one study investigated how long certain albums stay in the Billboard charts [33] while another showed which songs maintain their rankings on music download and streaming platforms [34].

Survival analysis is a tool that can take missing data into account. It therefore holds an advantage over other longitudinal analytical methods for investigating the present research question. In comparison to other methods like regression analysis, the dependent variable is not a continuous variable but instead represents the time unit until an event occurs. Usually, an event is a binary item that gives information about whether something happened or not. The other important variable is the so-called survival time which refers to the time an individual "survived" (participated without experiencing the event) during the follow-up time (a detailed description of this method can be found in [31]). An example related to the aim of the current study could be the time interval between which a student plays in a band before they decide to stop playing in the band. The event variable is dropping out of the band, while the survival time is the time during which the student played actively in the band until they stopped. The data is censored if the student did not drop out by the time the investigation ends. In this case, we cannot say whether the event ever occurs since it might happen after the time of investigation.

The present longitudinal study followed children aged between 10 and 17 years from British and German secondary schools, using survival analysis models to provide answers to the three primary research questions: 1. How many students engage in musical activities during adolescence? 2. Who will stop making music and when? 3. Which factors can explain who will survive puberty without giving up music making? For the second research question we will take into account individual difference factors. The previously discussed literature has shown that demographic variables such as age and gender are related to musical drop out. Individuals' personality, cognitive, or affective abilities are also considered to be influential factors [18], with intelligence strongly associated to continuing with music lessons ($r$ = .35-.36 [29]). From the comprehensive set of tests and questionnaires that are part of the test battery used in the longitudinal study, we decided to include age, gender, intelligence, and personality on account of these factors being highlighted as important within the literature. Additionally, we chose to use factors particularly related to music that are a central part of the longitudinal study. Taking into account music-related factors such as musical home environment, the type of instrument that is played and attitudes towards music, like the theory of musicality.

## 2. Method

The Goldsmiths Research Ethics & Integrity Sub-Committee approved the study (reference number: EA1230). Consent of the students' parents was obtained in written form. The data comes from a longitudinal project on the development of musical abilities. The project started with a pilot year of data collection in 2015, has grown substantially since and is still on-going. Every year, pupils complete a battery of musical listening tests and self-report questionnaires on psycho-social skills, attitudes, and leisure activities. The study does not include a music intervention, and there is no focus on a specific music genre or style. This study contains human research participant data. The subjects were secondary school students that shared sensitive data. Schools and German education ministries had us agree on data protection which is why we can only share data upon request (please contact Goldsmiths' Department of Psychology for this purpose, psychology@gold.ac.uk).

### 2.1 Participants

A total of 3,303 individuals participated in the longitudinal study between the years 2015 and 2019. They were recruited from four secondary schools in the UK and eight secondary schools in Germany (60.1% of the sample). All of them participated in the study at least once and only while they were between 10 and 17 years old. Out of these, 962 individuals participated twice,

**Table 1. Number of pupils per age (participants that participated multiple times were counted each time).**

| Age | 10 | 11 | 12 | 13 | 14 | 15 | 16 | 17 |
|-----|-----|------|------|-----|-----|-----|-----|-----|
| *N* | 963 | 1386 | 1211 | 615 | 618 | 376 | 266 | 147 |

55 three times, 47 four times and 7 five times in the longitudinal study. 60% identified as female, 35% as male, 2% as other, and 3% indicated that they did not want to disclose gender information. Table 1 shows how many pupils participated per age, reflecting the preponderance of participants between 10 and 12 years.

## 2.2 Procedure

Researchers visit schools each year at around the same time, to test pupils in their natural school setting. Students are allocated anonymous IDs when tested for the first time and are only identified by their ID upon retesting in subsequent years. Participants are tested in class groups, usually in the school's computer labs during normal school hours, seated in front of a computer with attached headphones (Behringer HPM1000). Where this is not possible, participants are given tablet computers with attached headphones for testing in their classrooms. Participants are instructed to work at their own pace through the online test battery. A testing session takes about 90 minutes.

## 2.3 Measurements

The measures described in this section are only those related to the current research questions. The longitudinal study features several additional tests and questionnaires that are listed in the study by Müllensiefen and colleagues [35].

**2.3.1 Concurrent musical activities.** The concurrent musical activities self-report inventory (CCM) assesses the degree of musical activity and active music making in adolescents over the past three months and therefore provides a snapshot of musical activity that is different to measures of musical background and expertise that cover longer periods or the entire lifetime (e.g., the Goldsmiths Musical Sophistication Index or Gold-MSI, see below). The CCM was developed on the basis of data on the range of musical activities of secondary school students [35] and consists of five binary items which assess engagement in musical activities (i.e. music making in orchestra/band, making music with friends or at special occasions, and currently receiving group or individual music instruction) as well as two rating scale items which gather information on the current amount of musical practice per day and the current total amount of music making per week (including practice, rehearsals, lessons, etc.). Only the 5 binary items are used as event variables in the context of survival analysis modelling in this paper.

**2.3.2 Musical sophistication.** Musical sophistication was assessed on the dimensions of the Goldsmiths Musical Sophistication Index (Gold-MSI [36]). Of particular interest for this study were the items asking for the musical instrument that students played and the age when they starting playing an instrument or singing. S1 Table lists all instruments that students indicated they were playing the most when they completed the battery for the first time.

**2.3.3 Personality.** The Big Five personality traits, namely Openness, Conscientiousness, Extraversion, Agreeableness, and Emotional Stability were assessed using a version of the Ten Item Personality Inventory (TIPI [37]) that was adapted for children from 10 years of age by Müllensiefen and colleagues [35]. Participants were asked to indicate on 7-point Likert scales how much they identified with certain attributes that were presented to describe a trait. All items and descriptive statistics of the TIPI can be found in S2 Table in the Appendix.

**2.3.4 Intelligence.** We used the MyIQ test [38], a computerized adaptive matrix reasoning test to assess general intelligence. Each item shows a matrix with nine spatial windows; eight windows contain visual patterns, and the participant is required to select a missing pattern to fit the ninth window. A total of eight potential choices are shown on each trial. Participants have a maximum of two minutes to answer each individual item. Within the longitudinal test battery, the MyIQ test was limited to eight trials where items were chosen according to an adaptive procedure following the approach by Müllensiefen and colleagues [35].

**2.3.5 Musical Home Environment (MHE).** To assess the level of musical activity in children's homes we constructed a new self-report inventory called *Musical Home Environment* (MHE) on the basis of data collected for the first year of data collection of the longitudinal project [36]. Data was collected for 31 binary items and four ratings scale items. Items asked about the musical activities of other people (e.g., parents, siblings) living in the same home household, for common musical activities at home, parents' encouragement and support with musical practice, the presence of a range of musical playback devices, and the attendance of musical events with family members. A Rasch item response model was computed from all binary items but revealed many empirical dependencies between items according to non-parametric tests (*t1*, *t11*) of item independence implemented in the R package eRm [39]. Only a set of four binary items passed these tests (i.e., Mother/Father playing an instrument or singing in a choir). A second Rasch model using only these four binary items was computed and this model passed all tests for Rasch model assumptions (i.e., unidimensionality, homogeneity, local item independence, subgroup invariance). Person (theta) scores were computed from this model and entered a subsequent PCA analysis together with the four rating scale variables. In addition to the theta scores derived from the binary items, only two rating scale items ('during holiday times my parents. . . encourage me to make music / support me practicing my instrument') had component loadings > 0.5 and hence were used in a final PCA model that explained 62% of the total item variance. Final scores of the MHE measure were computed as person scores from this PCA model.

**2.3.6 Theory of Musicality (TOM).** The Theories of Musicality (TOM) questionnaire [36] was used to assess pupils' attitudes towards the development of musical abilities [40]. The version of the TOM that was used for this analysis was modelled after the CNNAQ-2 [41] an inventory assessing self-theories in sports. The TOM measures have a hierarchical structure and comprises 12 items loading onto two main subscales. The Entity subscale measures whether participants believe that musical ability is unchangeable and a gift, while the Incremental Learning subscale assesses to what degree musical ability is believed to develop through learning.

## 2.4 Statistical analysis

The purpose of this study is to examine whether or not a pupil stayed musically active during his/her time in the longitudinal study. Survival analysis can provide an informative view of the survival over time, or—in the present case—indicate how long pupils stay musically active. The corresponding hazard rate indicates the likelihood of the occurrence of a certain event at a certain time–in our case dropping out of a musical activity at a certain age.

The binary dependent variable (dropout event / no dropout event) was computed from the binary items of the CCM measure. First, a *musical activity* variable was computed to indicate if a pupil engaged in any of the five possible musical activities (in an orchestra, with friends, at special occasions, as part of a group, or in individual music lessons). If a pupil stated that they engaged in one or more of the five activities the value 1 was coded for the musical activity variable, indicating that there was a musical activity, if not, a 0 was coded to indicate no musical

activity. Using this variable, a second variable of interest, namely *drop out*, was computed to indicate whether a participant had previously engaged but then stopped all musical activity at a certain time. The event was coded whenever a person indicated no musical activities (*musical activity* variable = 0) in a year that followed a year where they were musically active (*musical activity* variable = 1). If no activity followed a year with an activity the value 1 was assigned to the *drop out* variable indicating that the participant dropped out of their musical activity, otherwise assigning a value of 0. The time variable of interest was the age of the children, ranging from 10 to 17 years.

Survival analyses in the current study were carried out using the *survival* package [42] in R which uses the Kaplan-Meier estimate for single event survival. To take covariances into account, generalized linear (mixed) models were used with additional variables like personality, intelligence, MHE, and TOM as predictors to explain the binary drop out variable.

## 3. Results

### 3.1 Survival of musical inactivity

First, the complete sample was used, which included cases that never engaged in any musical activities and therefore could not have dropped out. This sample was used to determine the proportion of students that could potentially drop out. Therefore, a survival analysis looking at the *musical activity* variable was calculated that took into account all pupils who ever participated in the longitudinal study. Survival time in this case indicates how much time passed before pupils started to engage in a musical activity. The data was left and right censored since many pupils joined the study at an older age or their participation in the study ended at a certain age. The resulting model indicates that participants have a 52% chance of not engaging in any musical activity at the age of 10, but most of the pupils are likely to report an engagement in a musical activity at some point before they turn 17, with the highest probabilities at the ages of 11 and 12 years. The trajectory of the survival can be seen in the left-hand graph of Fig 1.

The hazard rate, on the other hand, indicates that participants were most likely to engage in a musical activity between the ages of 12 and 17 years, as can be seen in the right-hand graph of Fig 1.

### 3.2 Dropping out of musical activities

For the following analyses we used the second variable *drop out* that was computed to indicate whether participants stopped engaging in musical activities. Only participants who indicated

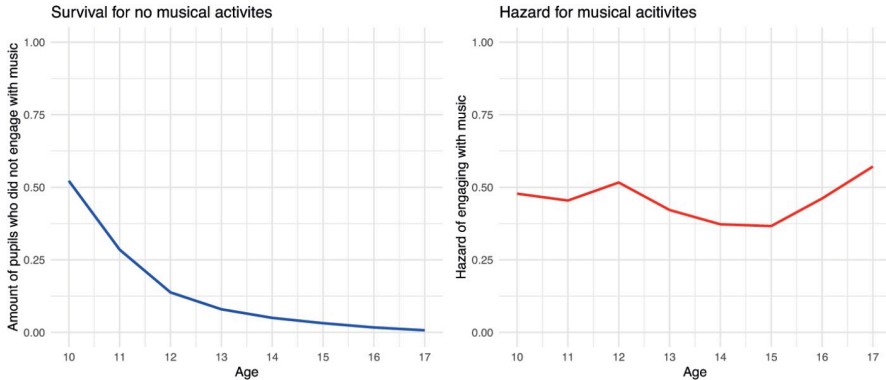

**Fig 1. Survival of musical inactivity and hazard of engaging with musical activities.**

**Table 2. Results of the survival analysis of musical drop out.**

| Age | Risk | Left | Censored | Hazard | Survival | *SE* |
|---|---|---|---|---|---|---|
| 10 | 2923 | 0 | 260 | 0.00 | 1.00 | 0.00 |
| 11 | 2663 | 54 | 600 | 0.02 | 0.98 | 0.003 |
| 12 | 2009 | 72 | 730 | 0.04 | 0.94 | 0.005 |
| 13 | 1207 | 26 | 308 | 0.02 | 0.92 | 0.006 |
| 14 | 873 | 16 | 192 | 0.02 | 0.91 | 0.007 |
| 15 | 665 | 37 | 235 | 0.06 | 0.86 | 0.011 |
| 16 | 393 | 61 | 167 | 0.16 | 0.72 | 0.018 |
| 17 | 165 | 53 | 112 | 0.32 | 0.49 | 0.029 |

that they engaged in a musical activity at one or more time points were included in these analyses. The results (see Table 2) show that almost half of all children keep engaging with musical activities until they turn 17 years old. The survival trajectory on the left-hand side of Fig 2 indicates that the number of musically engaged pupils decreases rapidly when they are 15 to 17 years old, while the hazard rate on the right-hand side of Fig 2 supports the finding that the most likely age to drop out of musical activities is between 15 and 17 with a peak at age 17.

### 3.3 Dropping out of musical activities by gender

In a next step, gender was used as a grouping variable, giving us two separate survival curves and hazard rates for female and male pupils. Fig 3 presents a visualization of the results and suggests that girls stay musically engaged for longer, as indicated by the green curve on the left-hand graph. The hazard rates on the right-hand graph show that boys are less likely to drop out of musical activity at the age of 16. The two graphs also show that there are no 17-year-old male pupils in this sample.

### 3.4 Dropping out of musical activities by instrument (piano and guitar)

Another factor that can help explain drop outs is the instrument that participants play, as indicated on the Gold-MSI self-report questionnaire. The two most frequently played instruments were piano and guitar. Therefore, we computed the survival curves and hazard rates for guitar and piano players that can be seen in the left-hand and right-hand graphs in Fig 4 respectively.

The survival curves show that the trajectory for staying musically engaged is roughly the same for years 10 to 16, with a decline that becomes steeper at age 16. When they turn 17,

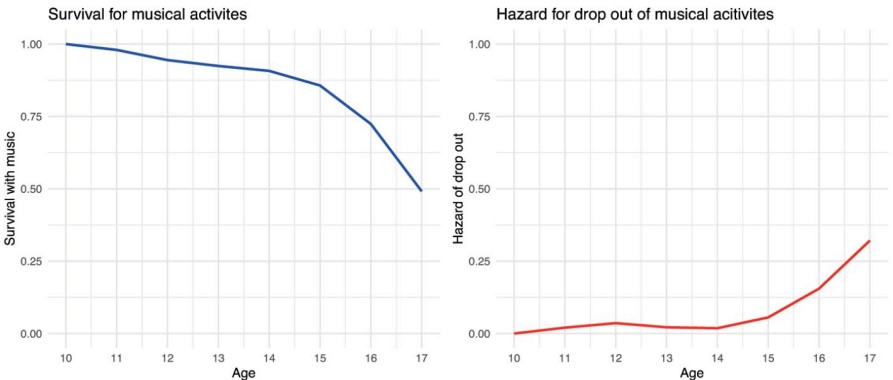

**Fig 2. Survival and hazard for musical drop out.**

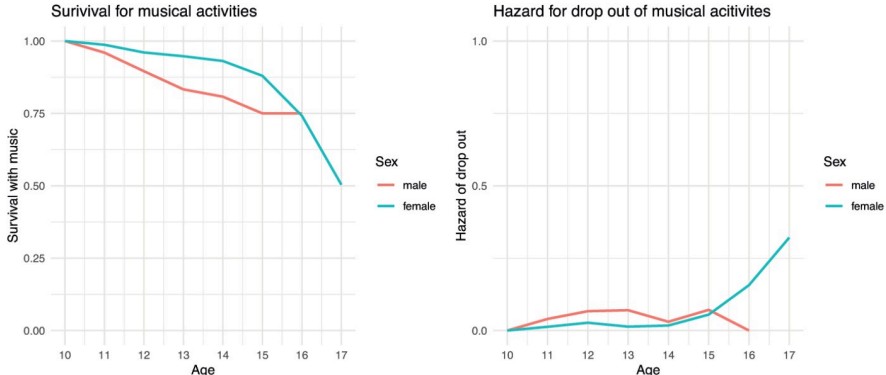

**Fig 3. Musical drop out for both genders.**

more piano players (as indicated by the green line) stay musically engaged than guitar players. The hazard rates support these findings, indicating that the possibility of dropping out of musical activities is highest for 17-year-old guitar players (see the orange line).

## 3.5 Influence of additional factors on dropping musical activities

To answer the question of which individual aspects influence whether pupils drop out of musical activities, several factors were used to predict the *drop out* variable. In a first step, the age of participants and their individual factors were taken into account. All students who participated at least two times in the longitudinal study and indicated they engaged in musical activities were included. Only students for whom observations were available from at least two time points were included in the sample. This sample ($n = 719$ with 1642 observations) was used to perform a hierarchical generalized mixed effects model with a Poisson distribution. Following the evidence from the empirical literature, predictor variables were included in five steps starting with the variable age, followed by the rather stable traits intelligence and personality. In the subsequent step variables that can exhibit change on shorter time scales (i.e., TOM and MHE) were entered into the model. The first step includes age ($\Delta R^2 = .09$), second step intelligence ($\Delta R^2 = .09$), third the five personality traits ($\Delta R^2 = .09$), thereafter the MHE ($\Delta R^2 = .14$), and lastly all previous variables, starting age of playing an instrument or singing and the TOM dimensions resulting in a full model with an $\Delta R^2$ of .17. $R^2$ values were computed using the *rsq*

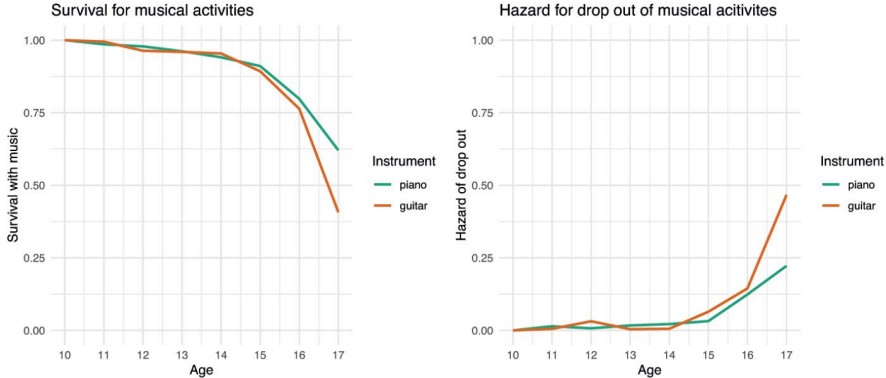

**Fig 4. Musical drop out for piano and guitar players.**

**Table 3. Results from the generalized linear mixed model as indicated by a back-fits test predicting musical drop out.**

| Predictors | Estimate | SE | Z value | p |
|---|---|---|---|---|
| Intercept | -6.06 | 0.94 | -6.435 | < .001*** |
| Age | 0.33 | 0.07 | 5.087 | < .001*** |
| Emotional stability | -0.19 | 0.11 | -1.708 | .09 |
| Musical home environment | -0.57 | 0.10 | -5.629 | < .001*** |
| $R^2$ = .14 | | | | |

$N$ = 719

***$p$ < .001.

package in R and refer to the complete model including mixed and random effects. In addition to the hierarchical construction of the regression model, a model backfitting approach was employed to identify the subset of predictors explaining most of the variance in the musical dropout variable, without overfitting the model. The Bayesian Information Criterion (BIC) was used as a model fit indicator and the minimum reduction in likelihood as threshold for the selection procedure using the glmer function in the *lme4* package in R. The resulting model only includes three predictors: age, emotional stability, and MHE, as can be seen in the model summary in Table 3.

When we look at age as a predictor in both generalized linear mixed models predicting drop out, results support the outcomes from the survival analyses with age as a significant predictor: The older participants become, the greater the risk of dropping out. The negative association between MHE and drop out indicates that a less strong musical environment of participants at home makes dropping out of musical activities more likely. Emotional stability was included as a predictor but seems less important as indicated by the *p*-value.

## 3.6 Influence of additional factors on dropping musical activities in a cross-sectional comparison

In a second step, the importance of all variables was tested in a cross-sectional comparison which included all individuals that engaged in music. One advantage of this model is that the data is modelled independent of age. Hence the focus is on predictors that are important across the adolescent period. Only data collected in the last year of participation from each participant who indicated they engaged in musical activities was included. With this cross-sectional dataset a hierarchical generalized linear model was performed using a binomial error distribution. The same hierarchical steps as before were applied (1st age, $\Delta R^2$ = .06; 2nd + intelligence, $\Delta R^2$ = .21; 3rd + personality, $\Delta R^2$ = .38; + MHE, $\Delta R^2$ = .52; 5th + TOM, $\Delta R^2$ = .54; 6th + starting age, $\Delta R^2$ = .69). Adjusted $R^2$ for the model (including random and fixed effects) was computed by applying a variance-function-based approach for generalized linear models using the R package *rsq*. Then, a stepwise regression analysis with a backward selection method, and the corrected version of the Akaike Information Criterion was applied to identify the best model. The variables of the model and their coefficients are given in Table 4.

The model shows that in addition to the musical home environment, theory of musicality and the personality trait conscientiousness are significant predictors. Age is not included as a predictor in the model constructed from the cross-sectional sample. The positive sign of the coefficient for TOM indicates that the more participants think of music as an entity, and the more conscientious they are, the more likely they are to drop out of musical activities. In accordance with previous analyses, the strongest predictor was MHE.

**Table 4. Results from the generalized linear model as indicated by a stepwise regression selection method predicting musical drop out.**

| Predictors | Estimate | SE | Z value | p |
|---|---|---|---|---|
| Intercept | -3.38 | 1.05 | -3.232 | .001** |
| Musical home environment | -0.71 | 0.12 | -6.093 | < .001*** |
| TOM entity | 0.50 | 0.22 | 2.276 | .02* |
| Conscientiousness | 0.30 | 0.13 | 2.229 | .03* |
| Emotional stability | -0.23 | 0.15 | -1.504 | 0.13 |

$N$ = 404

***$p$ < .001

**$p$ < .01

*$p$ < .05.

$R^2$ = .55, accuracy = .83.

## 4. Discussion

The results from the survival analyses show that about 50% of those who engage in musical activities drop out by the time they turn 17 years old which is in line with the findings by Theorell and colleagues [28]. The age at which most children show musical drop out is not surprising since it is around this age that they go through puberty. The rates of musical drop out start to increase when children turn 15 years old, which is about two years after the start of puberty, where many physiological and social changes happen in their lives [43]. These changes associated with puberty seem to be a likely explanation for the trajectories modelled from the data. Furthermore, this is around the age where students might have tried out various leisure time activities and may settle for a specific one (e.g., athletics) while at the same time neglecting others (e.g., music).

Although female pupils are more likely to stay musically engaged until they are 15 years old, the decrease of their engagement is even more drastic when they are 15 and 16 years old whereas the curve of the male pupils appears to be more akin to a linear downward trend starting at the age of 10. These results should be regarded with caution, however, since fewer boys than girls participated in the study and there were no boys over the age of 16. Nevertheless, the survival curves show that among younger pupils (ages 11 to 13) the hazard of dropping out of musical activities is higher for boys than girls which is in line with existing research [28]. In the past, findings [4] have pointed out that there are gender differences when it comes to musical activities, with one explanation being that boys sometimes regard musical activities as "feminine" and therefore find them less desirable as a leisure time activity.

We only found slight differences between piano and guitar players, with guitar players more likely to drop out of musical activities than piano players when they are between 16 and 17 years old which slightly contradicts previous findings from Theorell and colleagues [28]. One explanation for the present finding could be that guitar players are more likely to feel that they have reached their goals in terms of musical learning at an earlier stage since it an easier instrument to achieve basic levels of playing [3].

Age seems to play a very important role when it comes to musical survival. Clear directions of all trajectories can be seen and in the predictive mixed model as well as the survival models age was a significant factor. The older children grow, the more important other leisure activities become [44] and it is to be expected that musical drop out will increase with age. Although the importance of music is at its height at this age [45] the actual engagement with musical activities like participating in music lessons or group music making has to compete with other

activities. Another explanation could be that playing certain types of instruments are not considered to fit with the 'norm' which could be due to socialisation or even peer pressure [3].

Another important factor for sustained instrumental engagement reported in the literature is the age at which children start playing an instrument or singing [28]. Yet, we did not find starting age to be a significant predictor in the current study which could be due to the explanatory power of the MHE which is correlated with starting age ($r = 0.14$, $p < .001$) and might explain in more detail how children are introduced to music during their childhood. MHE takes into account not only the parental musical activity but also to what degree parents encourage and support musical learning and practice at home. This measure can therefore be considered a proxy for the importance that parents assign to musical activity and this is possibly also related to the age at which parents offer their children the opportunity to start musical lessons.

Individual factors like personality seem to explain the likelihood of musical drop out to some degree. The finding that more conscientious pupils are more likely to drop their musical activities might appear surprising, since studies indicated that conscientiousness is connected to engagement in music lessons [46]. However, it could be argued that more conscientious pupils focus on school, learning and their academic achievement and are therefore more likely to drop out of leisure activities such as music in favour of their studies. Another explanation could be that a high level of conscientiousness might lead to not feeling successful in performing on the instrument of choice and therefore dropping music.

The musical environment at home is another important factor that influences musical engagement [47]. Parents, the presence of musical instruments at home, the frequency of musical activities and even listening to music as a family can contribute to a more motivated and longer lasting engagement with music [47]. This supports our findings which showed that drop out is more likely in pupils that have a less musical home environment.

## 4.1 Limitations

First, despite contributing numerous findings, the longitudinal method used for this study has its limitations. First, the sample is skewed towards younger participants. This could be explained by the fact that data collection at some schools included only first- and second-year students. However, because survival analysis takes censored data into account, the results can still provide valuable insights. Nevertheless, future studies might want to take a closer look at older children to investigate the drop out of 16 and 17-years-olds.

Second, only a proportion of pupils participated more than once in our study. Which is why the current investigation should be rather described as a study on students across ages than a longitudinal study. To reveal more comprehensive trajectories participants should be tested for a longer time. The project is only at its beginning and we feel confident that we will have more reliable longitudinal data given a few more years. Additionally, we would recommend that future studies might want to choose shorter intervals between repeated test sessions to help with consistency of participation and enable a more detailed view of development.

Third, the drop out variable was computed based on the items of the concurrent musical activity inventory. Since the focus of the whole longitudinal project is on the development of musical abilities, this variable was sufficient for our purposes, but future studies might want to include items into their batteries that focus on the actual drop out event and incorporate questions on reasons for quitting.

Fourth, our data come from all kinds of schools (public, private, boarding etc.) which means the schools' curricula and extraordinary school activities do not necessarily include music lessons or classes. Therefore, some grouping variables might be not as representative as

we would hope for. Samples from schools with a focus on music might provide a wider range of instrument players and more state/public schools would help to get more representative demographics. Previous studies already indicated that the school type has a significant impact on musical development [28]. Therefore, future studies that want to focus on gender, instrument players, or other grouping factors should carefully select the participating schools during the recruiting process and maybe include a variable that accounts for musical opportunities at each school.

Fifth, the personality measurement (TIPI) is a brief measurement of a very complex disposition which might be regarded as insufficient for a detailed analysis of one's personality. The length of the questionnaire is, however, a huge advantage when included as part of a long test battery like the one used for the present study. Furthermore, validation studies of the TIPI show acceptable metrics [48]. Still, future studies looking into how personality and musical activities are correlated might want to use a more detailed measurement.

Lastly, the aim of this study was to investigate how music related factors influence engagement with musical activities, and therefore only two other factors, intelligence and personality, were used in addition. Future studies might want to incorporate other factors and measurements, such as learning behaviour or school engagement, which might give insights into levels of motivation and previous experience with learning strategies [49]. Since in many countries (e.g., the USA) sports activities (e.g., athletic programs) can be important for obtaining university scholarships and later employment, future studies might want to take into account competing activities like sports or drama that might explain why students drop out of musical activities in favor of others. Furthermore, taking genetic factors into account which could explain relationships between musical abilities, musical engagement, and intelligence using genetic pleiotropy seems to be a fruitful explanatory approach and should be explored in more detail by future research taking into account studies like the one by Mosing and colleagues [50]. These variables could account for some of the unexplained variance of the models presented in this study.

### 4.2 Conclusion

If we want to help children and adolescents stay musically engaged, the results of this study suggest that we should focus on older children, especially those aged 15 years and older. Music teachers and instructors should advise parents to create a supporting musical home environment and to check in with their children frequently regarding their engagement with musical activities. Also, explaining how musical development takes place and helping children to understand that musicality is not an unchangeable entity but an ability that can be improved with practice should be beneficial for fostering ongoing musical engagement. Moreover, an incremental view of musicality seems to be important for a positive attitude towards learning in general and can help with academic achievement [34].

In sum, the current study has revealed factors that contribute to continued active engagement with music during adolescence and showed that the risk for dropping out of musical activities is highest between 15 and 17 years. These insights may help to prevent subsequent generations of adolescents asking themselves the question: Why did I ever stop making music?

### Supporting information

**S1 Table. Self-reported most played instruments of first-time participants.**
(DOCX)

**S2 Table. Ten Item Personality Inventory (TIPI; Gosling, et al., 2003)–extended.** $N = 3,915$, all items were measured on 7-point Likert scales, ranging from 1 (Disagree strongly) to 7 (Agree strongly), adjectives in italic were added to the original scale.
(DOCX)

## Acknowledgments

We are extremely grateful for the logistic and organisational support that this project has received from Queen Anne's School and all UK and German schools that participated and still participate in the LongGold project. We would like to thank Chloe Stacey MacGregor for proofreading.

## Author Contributions

**Conceptualization:** Nicolas Ruth.

**Data curation:** Nicolas Ruth.

**Formal analysis:** Nicolas Ruth.

**Funding acquisition:** Nicolas Ruth, Daniel Müllensiefen.

**Investigation:** Nicolas Ruth.

**Methodology:** Nicolas Ruth.

**Project administration:** Nicolas Ruth, Daniel Müllensiefen.

**Resources:** Daniel Müllensiefen.

**Visualization:** Nicolas Ruth.

**Writing – original draft:** Nicolas Ruth.

**Writing – review & editing:** Daniel Müllensiefen.

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
