## [Decision Letter · Decision Letter 0]

27 May 2021

PONE-D-21-12414

Survival of musical activities. When do young people stop making music?

PLOS ONE

Dear Dr. Ruth,

Thank you for submitting your manuscript to PLOS ONE. After careful consideration, we feel that it has merit but does not fully meet PLOS ONE’s publication criteria as it currently stands. Therefore, we invite you to submit a revised version of the manuscript that addresses the points raised during the review process.

We look forward to receiving your revised manuscript.

Kind regards,

Andrea D. Warner-Czyz, Ph.D.

Academic Editor

PLOS ONE

Journal Requirements:

2. PLOS ONE does not copy edit accepted manuscripts (https://journals.plos.org/plosone/s/criteria-for-publication#loc-5). To that effect, please ensure that your submission is free of typos and grammatical errors.

Reviewers' comments:

Reviewer's Responses to Questions

**Comments to the Author**

1. Is the manuscript technically sound, and do the data support the conclusions?

Reviewer #1: Yes

Reviewer #2: Yes

2. Has the statistical analysis been performed appropriately and rigorously? 

Reviewer #1: Yes

Reviewer #2: Yes

3. Have the authors made all data underlying the findings in their manuscript fully available?

Reviewer #1: No

Reviewer #2: Yes

4. Is the manuscript presented in an intelligible fashion and written in standard English?

Reviewer #1: No

Reviewer #2: Yes

5. Review Comments to the Author

Reviewer #1: With shrinking space for art education at school in most Western countries it is becoming increasingly important to examine factors that stimulate sustenance of art habits for life. The present contribution is important from this point of view.

In the art research field it is very difficult to find all the relevant literature studies published in all kinds of scientific journals. I have with coworkers done one of the few studies examining factors that make children continue music practicing as adults but the authors have obviously not seen it, maybe because it was published in a paediatric journal in which one would not expect to find such an article. However, it examines many of the questions that the authors are addressing so it is relevant (Acta Paediatrica 104 :274-284, 2015). It was a quasi-prospective study based upon a large Swedish twin cohort (with random exclusion of one member of each twin pair – to equate it with a normal population study) with 3820 participants aged 27-54 who were asked to reconstruct number of training hours as well as a number of other relevant conditions during different age periods. Some of the conclusions are similar to the ones that the authors are making here, despite the fact that widely different statistical analyses were performed. 70% had had some exposure to music teaching added to the regular school curriculum, and almost exactly as the authors report in the present study only half of those subjects continued active music practice as adults. The likelihood of sustained music practice up to adult years decreased with start of extra music. The calculation in the Swedish study was that for every year of delayed start of extra music between age 6 and 17 there was a 7% decrease in likelihood that the participant would continue music practice up to adult years. It would be interesting if the authors could do a similar calculation in the present study. It is an important perspective because the Swedish findings mean that early start is very important.

As in the present study, there were fewer boys who started extra music, but on the other hand those boys who did were more likely than girls to keep their habit. This is similar to the present study. As in the present study genre was important as well as instrument – but the findings for piano and guitar were widely diverging in the two studies. Music and other cultural activities in the families were important survival factors as were special public music schools (which were attended by few and probably talented) but also community cultural schools for leisure art learning (much larger numbers) increased survival slightly.

In the Swedish study in which sampling represented a much longer period it was also possible to examine a cohort effect – participants continued active music practice as adults to a much smaller extent if they went to school during the 2000 :s than if they did so during the late 1970 :s.

I mention this because it would be really nice if these two studies although they have different research questions and are quite different in design could be compared in the discussion. It would also be interesting if the authors could refer to the Swedish twin studies (Ullén has been PI) which have explored the role of genetic factors which the authors also discuss. For instance pleiotropy may play a role in several of these relationships.

The authors mention in the discussion that they feel that the cross-sectional analysis has advantages. But it also has disadvantages : We do not know whether a high TOM score is secondary to a failure in music practice or whether it is a true predictor (rationalization retrospectively : I now understand that I failed because musical talent cannot be taught). Their discussion around conscientiousness is interesting, but the cross-sectional design makes it impossible to know whether it is causal. The family culture factor however seems very established and is important also in the Swedish study.

However, in general this is an important study.

Reviewer #2: This paper included a survival analysis to document ages at which student participation in music declines. The authors utilized a longitudinal study of students 10-17 years of age and found that home musical environment was an important factor in whether students persisted in the latter ages. While the authors state other papers have provided a rationale to "why" students may choose not to persist in continuing music performance and stated that may be in forthcoming studies, I found myself wanting to link the ages to the rationale for the choice to forego continuing in music.

Specific concerns in the paper include the use of "(e.g., [citation])" when the "e.g." felt unwarranted

There are several typographical errors in the paper that can be remedied quickly

P 9, L 197: Write out the citation for #34

P10, L 223: revise "Table6Table 5"

P 12, L 259: remove redundant "scale"

P 15, 325: revise "that boys have a are less likely"

Discussion: there are other likely rationale besides puberty for decisions not to persist in music. In the US (understood this study did not originate there), athletics are in higher demand for time as well as employment, and often conflict with musical requirements. Many studies find those conflicts are a leading cause is attrition in music programs. Is there a way to connect your work with those findings? I think that combination would truly show the decline is more than anecdotal, as your data show.

None of the Figures have captions which makes it hard for the reader to connect the writing to the information (aside from very small print). This is the most significant need to address.

6. PLOS authors have the option to publish the peer review history of their article (what does this mean?). If published, this will include your full peer review and any attached files.

Reviewer #1: **Yes: **Tores Theorell

Reviewer #2: No

---

## [Author Response · Author response to Decision Letter 0]

20 Jul 2021

Dear editor and reviewers,

we wrote a rebuttal to the reviews and uploaded it with the new manuscript. 

There are indeed ethical or legal restrictions to sharing our data publicly, we explained them in detail now.

The revisions greatly improved the manuscript and we hope that it is fitting for your journal at this stage.

All the best

Nicolas Ruth & Daniel Müllensiefen

---

## [Decision Letter · Decision Letter 1]

21 Sep 2021

PONE-D-21-12414R1Survival of musical activities. When do young people stop making music?PLOS ONE

Dear Dr. Ruth,

Thank you for submitting your manuscript to PLOS ONE. After careful consideration, we feel that it has merit but does not fully meet PLOS ONE’s publication criteria as it currently stands. Therefore, we invite you to submit a revised version of the manuscript that addresses the points raised during the review process.

I agree with the reviewers that the authors improved this manuscript considerably since its initial submission. However, the paper would benefit from a few minor changes, as recommended by Reviewer 2.==============================

We look forward to receiving your revised manuscript.

Kind regards,

Andrea D. Warner-Czyz, Ph.D.

Academic Editor

PLOS ONE

Journal Requirements:

Additional Editor Comments (if provided):

Reviewers' comments:

Reviewer's Responses to Questions

**Comments to the Author**

1. If the authors have adequately addressed your comments raised in a previous round of review and you feel that this manuscript is now acceptable for publication, you may indicate that here to bypass the “Comments to the Author” section, enter your conflict of interest statement in the “Confidential to Editor” section, and submit your "Accept" recommendation.

Reviewer #1: All comments have been addressed

Reviewer #2: (No Response)

2. Is the manuscript technically sound, and do the data support the conclusions?

Reviewer #1: Yes

Reviewer #2: Partly

3. Has the statistical analysis been performed appropriately and rigorously? 

Reviewer #1: Yes

Reviewer #2: Yes

4. Have the authors made all data underlying the findings in their manuscript fully available?

Reviewer #1: No

Reviewer #2: No

5. Is the manuscript presented in an intelligible fashion and written in standard English?

Reviewer #1: Yes

Reviewer #2: Yes

6. Review Comments to the Author

Reviewer #1: The authors have adequately dealt with the comments

I cannot find any information regarding accessibility of data for other researchers but I may not have looked for it sufficiently. However, there is reason to believe that the authors will give access to their data

Reviewer #2: The revision is much improved and I thank the authors for their time in addressing the concerns. My remaining issues are minor and can be addressed quickly, I believe. They are summarized below:

P 11, L234: "Table6Table5" is still included in the sentence and should be edited to reflect the correct reference

P20, L432: you state "..several studies [4]" and cite only the one. Please reflect multiple studies.

P21, L453: suggest moving the r and p values after "with starting age" to make it easier for the reader.

P21, second paragraph: the authors argue that conscientious pupils focus on school and drop leisure activities, but it could also be argued that a high level of conscientiousness may lead to dropping music due to not feeling successful in performing on the instrument of choice

P22, 480: if only a small proportion of students participated more than once, can you truly call it a "longitudinal study". Rather, it is more of a sample across ages. Perhaps remove "longitudinal" since that could be misleading

P23, LL516-518: evaluating genetic factors to explain relationships is a large leap to take in your study, given that single added citation. Suggest removing this.

7. PLOS authors have the option to publish the peer review history of their article (what does this mean?). If published, this will include your full peer review and any attached files.

Reviewer #1: **Yes: **Töres Theorell

Reviewer #2: No

---

## [Author Response · Author response to Decision Letter 1]

23 Sep 2021

We uploaded a reply to the reviews as a seperate document.

---

## [Decision Letter · Decision Letter 2]

13 Oct 2021

Survival of musical activities. When do young people stop making music?

PONE-D-21-12414R2

Dear Dr. Ruth,

We’re pleased to inform you that your manuscript has been judged scientifically suitable for publication and will be formally accepted for publication once it meets all outstanding technical requirements.

Kind regards,

Andrea D. Warner-Czyz, Ph.D.

Academic Editor

PLOS ONE

Additional Editor Comments (optional):

Reviewers' comments:

Reviewer's Responses to Questions

**Comments to the Author**

1. If the authors have adequately addressed your comments raised in a previous round of review and you feel that this manuscript is now acceptable for publication, you may indicate that here to bypass the “Comments to the Author” section, enter your conflict of interest statement in the “Confidential to Editor” section, and submit your "Accept" recommendation.

Reviewer #1: All comments have been addressed

Reviewer #2: All comments have been addressed

2. Is the manuscript technically sound, and do the data support the conclusions?

Reviewer #1: (No Response)

Reviewer #2: Yes

3. Has the statistical analysis been performed appropriately and rigorously? 

Reviewer #1: (No Response)

Reviewer #2: Yes

4. Have the authors made all data underlying the findings in their manuscript fully available?

Reviewer #1: (No Response)

Reviewer #2: No

5. Is the manuscript presented in an intelligible fashion and written in standard English?

Reviewer #1: (No Response)

Reviewer #2: Yes

6. Review Comments to the Author

Reviewer #1: (No Response)

Reviewer #2: All concerns submitted in previous reviews have now been addressed by the authors. I thank them for their clarifications.

7. PLOS authors have the option to publish the peer review history of their article (what does this mean?). If published, this will include your full peer review and any attached files.

Reviewer #1: **Yes: **Tores Theorell

Reviewer #2: No

---

## [Editor Report · Acceptance letter]

28 Oct 2021

PONE-D-21-12414R2 

Survival of musical activities. When do young people stop making music? 

Dear Dr. Ruth:

I'm pleased to inform you that your manuscript has been deemed suitable for publication in PLOS ONE. Congratulations! Your manuscript is now with our production department. 

Kind regards, 

on behalf of

Dr. Andrea D. Warner-Czyz 

Academic Editor

PLOS ONE